# Discriminator-Guided Model-Based Offline Imitation Learning

**Wenjia Zhang[1], Haoran Xu[2], Haoyi Niu[1], Peng Cheng[3],**
**Ming Li[1], Heming Zhang[1], Guyue Zhou[1], Xianyuan Zhan[1,4*]**
[1] Tsinghua University, Beijing, China
[2] JD Technology, Beijing, China
[3] Beijing Jiaotong University, Beijing, China
[4] Shanghai AI Laboratory, Shanghai, China
`zhangwj18@mails.tsinghua.edu.cn`
`{ryanxhr,t6.da.thu,liming18739796090}@gmail.com`
`20112041@bjtu.edu.cn`
`hmz@mail.tsinghua.edu.cn`
`{zhouguyue,zhanxianyuan}@air.tsinghua.edu.cn`

**Abstract:** Offline imitation learning (IL) is a powerful method to solve decision-making problems from expert demonstrations without reward labels. Existing offline IL methods suffer from severe performance degeneration under limited expert data. Including a learned dynamics model can potentially improve the state-action space coverage of expert data, however, it also faces challenging issues like model approximation/generalization errors and suboptimality of rollout data. In this paper, we propose the Discriminator-guided Model-based offline Imitation Learning (DMIL) framework, which introduces a discriminator to simultaneously distinguish the dynamics correctness and suboptimality of model rollout data against real expert demonstrations. DMIL adopts a novel cooperative-yet-adversarial learning strategy, which uses the discriminator to guide and couple the learning process of the policy and dynamics model, resulting in improved model performance and robustness. Our framework can also be extended to the case when demonstrations contain a large proportion of suboptimal data. Experimental results show that DMIL and its extension achieve superior performance and robustness compared to state-of-the-art offline IL methods under small datasets.

**Keywords:** Offline Imitation Learning, Model-based Learning, Sample Efficiency

## 1 Introduction

Offline imitation learning (IL) that trains a policy from expert demonstrations without additional online environment interactions has become an attractive solution for many real-world decision-making applications, such as robotic manipulation [1] and autonomous driving [2, 3], etc. It bypasses several major obstacles in practice, such as the difficult reward function design [4] as in reinforcement learning (RL) approaches, and the requirement of simulation or real-world system interactions during model training as in online IL methods [5, 6, 7, 8, 9], which can be costly or dangerous.

Despite these desirable features, the performance of offline IL methods heavily depends on the size and quality of demonstration data. Due to its supervised learning nature, learning an IL policy in parts of the state space not covered by expert data could make arbitrary mistakes, which leads to severe compounding errors. This phenomenon, called *covariate shift* [10, 11, 12], is a core issue in IL and greatly hurts the policy generalization capability. In practice, collecting a large number of expert demonstrations can be costly or infeasible. The reduction in data size coupled with the narrow expert data distribution can lead to limited state space coverage, causing poor policy performance. On the other hand, involving non-expert suboptimal offline demonstration data although can potentially

---

*Corresponding author.

6th Conference on Robot Learning (CoRL 2022), Auckland, New Zealand.

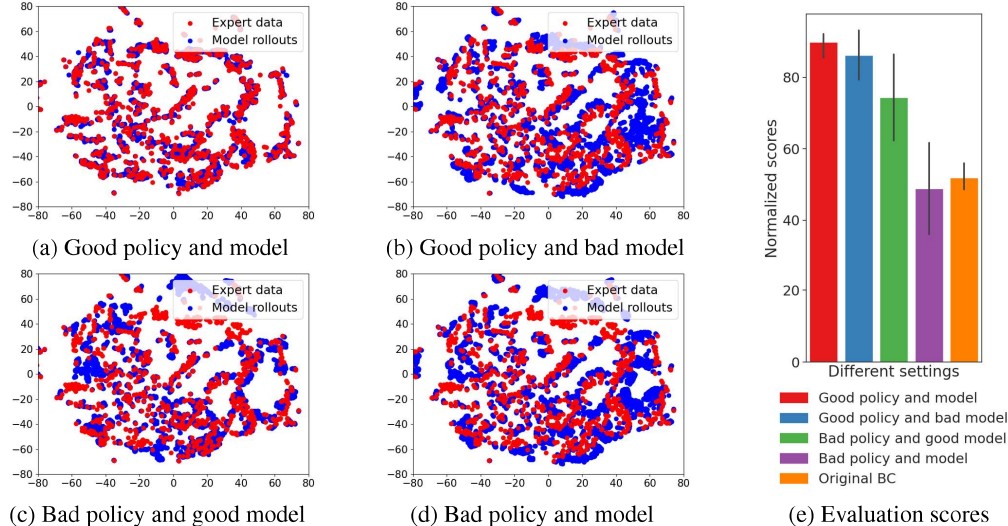

Figure 1: Empirical observations on the impacts of involving dynamics model rollouts in BC. (a)-(d) TSNE visualizations of expert data and dynamics model rollouts under different BC policies on MuJoCO Hopper task with only 20,000 expert data transitions (2% of the D4RL [29] Hopper-expert dataset). The good policy and dynamics model are two-layer MLPs with 256 hidden units, and are trained until convergence. The bad policy and model are trained with fewer steps, and the hidden layers of the latter are reduced to 128 units. It can be observed that model rollouts under well-learned policy and dynamics model align well with the expert data, while noticeable discrepancies are observed when the policy or the model is problematic. (e) shows the final performance of BC policy trained with 1:1 expert and model rollout data under the four cases in (a)-(d). It is found that under small expert datasets, including a dynamics model in many cases is beneficial, but the quality of rollout policy and dynamics model could have great impact on the final policy performance.

improve state-action space coverage, is shown in previous studies [13, 14] to result in reduced performance in traditional offline IL methods like behavior cloning (BC) [2]. Many of these problems can be alleviated in the online IL setting, either by interactively querying an expert to collect more data [5, 15, 16], or by resorting to inverse reinforcement learning (IRL) to learn a rewards function or match the state-action distribution induced by the expert policy [6, 8, 9, 17]. However, such treatments do not apply to the offline setting, since additional environment interaction is not possible. Moreover, utilizing additional suboptimal offline data through offline IRL approaches [13, 18] also shows inferior performance compared with online IRL counterpart methods, due to the involvement of offline RL sub-problems that is prone to training instability and bootstrapping error accumulation [19, 20]. Hence, the ability to leverage limited expert data for robust policy learning remains to be a key challenge for the successful real-world deployment of offline IL methods.

The sample efficiency requirement for offline IL methods reminds us of the success of model-based approaches in the online and offline RL domains [21, 22, 23, 24, 25]. Dynamics models learned from the data can greatly supplement the limited expert data to improve state-action space coverage, leading to potentially improved policy performance and generalizability [22, 25, 26, 27]. However, adopting a model-based approach in offline IL is still an underexplored area [26, 27, 28]. Many existing methods bear some limitations, such as requiring an additional suboptimal dataset [26] or a low-fidelity simulator [28] for training, or fully trusting the learned dynamics model [27]. The key challenges of introducing a learned dynamics model in IL policy learning is twofold (see Figure 1 for an empirical illustration): 1) the learned dynamics model has approximation/generalization errors, directly using model rollouts for imitation learning can be problematic; 2) using the learned policy as the rollout policy may generate suboptimal data, causing performance degeneration that similar to the case of learning with suboptimal data in IL [13, 20]. In model-based RL, the second problem is less severe, as the reward function can be used to distinguish the optimality of data. However, this is typically not possible in IL settings.

In this work, we develop a novel model-based offline IL framework to tackle the above challenges. We introduce a discriminator to simultaneously distinguish the dynamics discrepancy and suboptimality of the model rollout data against the real expert demonstrations. This gives rise to a special cooperative-

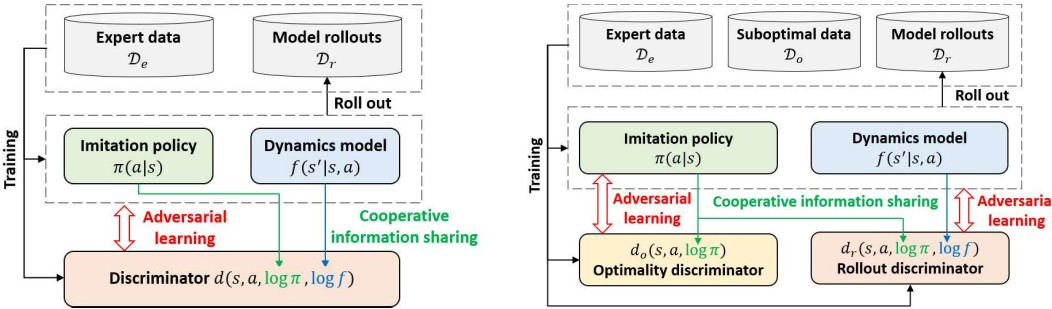

(a) DMIL: learning from expert demonstrations only   (b) D2MIL: learning from both expert and suboptimal data

Figure 2: Illustration of the proposed model-based offline IL framework DMIL and its extension D2MIL

yet-adversarial "three-party game". Both the dynamics model and the policy provide information as inputs to the discriminator, while also challenging it to establish worst-case error minimization. Under this design, the discriminator can use more information to make better judgment on the dynamics correctness and optimality of the rollout data, and the worst-case optimization scheme also substantially improves the robustness of all three models (policy, dynamics model and discriminator). Interestingly, we can show that this design leads to new IL policy and dynamics model learning objectives, where the outputs of the discriminator sever as weights in their original loss functions. Moreover, the resulting algorithm can be efficiently solved in a simple supervised learning manner, which avoids explicitly solving the complex min-max optimization problems as in adversarial learning [30, 31]. We thus term our algorithm Discriminator-guided Model-based Imitation Learning (DMIL). Our proposed framework can also be extended to the offline IL setting that involves limited expert and a larger proportion of unknown quality, potentially suboptimal data [13, 20]. This can be achieved by simply introducing the second discriminator to contrast the expert and suboptimal data, which we refer this variant as Dual-Discriminator guided Model-based Imitation Learning (D2MIL). Through extensive experiments on D4RL benchmarks [29] and real-world robotic tasks, we show that both DMIL and D2MIL achieve superior performance and robustness against state-of-the-art methods under small datasets. These promising results demonstrate the potential of adopting model-based learning in real-world offline IL applications under limited expert demonstrations.

## 2 Method

### 2.1 Problem Setting

We consider the fully observed Markov Decision Process (MDP) setting, which can be described as $\mathcal{M} = (\mathcal{S}, \mathcal{A}, P, d_0, r, \gamma)$, where $\mathcal{S}$ and $\mathcal{A}$ are the state and action space, respectively, $P(s'|s, a)$ is the transition probability, $d_0(s)$ is the initial state distribution, $r(s, a)$ is the reward function, and $\gamma \in [0, 1]$ is the discount factor. Under offline IL setting, we have an expert dataset $\mathcal{D}_e = \{(s_i, a_i, s'_i)\}_{i=1}^N$ collected from some expert policy $\pi_e$. Our goal is to learn a policy $\pi(a|s)$ to minimize its gap with the expert policy $\pi_e$. In the simplest case, behavior cloning (BC) trains the policy by minimizing the negative log-likelihood of the observed expert actions:

$$\min_\pi \mathcal{L}_\pi := \mathbb{E}_{(s,a)\sim\mathcal{D}_e}[-\log \pi(a|s)] \tag{1}$$

### 2.2 Discriminator-Guided Model-Based Imitation Learning (DMIL)

Traditional offline IL methods like BC suffer greatly from covariate shift under small expert datasets due to extremely sparse state space coverage of data. Our idea is to mitigate this issue by involving dynamics model rollouts while also carefully handling these potentially problematic data through the guidance of an additional discriminator in a coupled and cooperative-yet-adversarial learning process. Figure 2 provides an illustration of the proposed DMIL framework as well as its extension D2MIL.

**Incorporating the Dynamics Model.** Model-based approaches have been widely adopted in RL to improve sample efficiency and shows good performance and generalization ability in recent offline

RL studies [22, 24, 25]. In our work, we introduce a probabilistic dynamics model implemented using a neural network that outputs a Gaussian distribution over the difference between the current and next state, i.e., $f(s'|s,a) = \mathcal{N}(s + \mu_{\theta_f}(s,a), \Sigma_{\theta_f}(s,a))$, where $\mu_{\theta_f}(s,a)$ and $\Sigma_{\theta_f}(s,a)$ are the parameterized mean and diagonal covariance matrix. We predict the difference of states rather than the next states as it has been shown in past studies [21, 22] to yield better dynamics predictions. The dynamics model can be learned using the following maximum log-likelihood objective:

$$\min_f \mathcal{L}_f := \mathbb{E}_{(s,a,s') \sim \mathcal{D}_e}[-\log f(s'|s,a)] \tag{2}$$

**Cooperative-yet-Adversarial Learning Scheme.** Directly using the rollout data $\mathcal{D}_r$ generated by the learned BC policy $\pi$ and dynamics model $f$ in subsequent imitation learning can be problematic. Under small datasets, it is usually difficult to obtain an accurate dynamics model, and the rollouts from a less-well learned policy can be suboptimal compare with the true expert data. To solve this issue, we use a discriminator $d$ to measure dynamics discrepancy and suboptimality in rollout data $\mathcal{D}_r$. Moreover, we introduce a special cooperative-yet-adversarial learning scheme, and use the discriminator as a bridge to couple the learning process of $\pi$, $f$ and $d$. The key idea is to first include the element-wise loss information from both policy $\pi$ and dynamics model $f$ (i.e., $\log \pi$ and $\log f$) into the input of the discriminator (i.e., $d(s,a,\log \pi(a|s), \log f(s'|s,a))$) to establish cooperative information sharing. And then make $\pi$ and $f$ challenge $d$ to establish adversarial learning. This leads to a special learning objective for the discriminator $d$, which can be expressed as:

$$\min_d \max_{\pi,f} \mathcal{L}_d := \mathbb{E}_{(s,a,s') \sim \mathcal{D}_e}[-\log d(s,a,\log \pi(a|s), \log f(s'|s,a))] +$$
$$\mathbb{E}_{(s,a,s') \sim \mathcal{D}_r}[-\log(1 - d(s,a,\log \pi(a|s), \log f(s'|s,a)))] \tag{3}$$

This design has a number of attractive properties. First, element-wise loss information from $f$ and $\pi$ reflects the confidence of these models on the rollout data. Suppose $f$ and $\pi$ are well-learned, then they will assign high probabilities (large $\log \pi$ and $\log f$) on good rollouts with reasonable dynamics and expert-like samples. This can provide valuable information to facilitate the judgment of the discriminator. Second, the adversarial component forms a GAN-like problem [30], where $\pi$ and $f$ jointly serve as a generator to challenge the discriminator. This will force the discriminator to minimize the worst-case error [31, 32], which makes its robustness significantly improved. In return, a stronger $d$ can better guide the learning of $\pi$ and $f$ to further improve their performance and make better use of the generalization power of the dynamics model. Consequently, this cooperative-yet-adversarial learning scheme enables coupling among policy, dynamics model and discriminator, which can potentially lead to boosted performance for all three models.

**Loss Correction for Policy and Dynamics Model.** Jointly solving Eq.(3) together with minimization problems in Eq.(1) and (2) can be rather complex. As both $\pi$ and $f$ appear in the input of the discriminator, $d$ becomes a functional of $\pi$ and $f$ (i.e., function of a function). Eq.(3) is a functional min-max optimization problem, which is itself quite challenging to solve. Fortunately, based on calculus of variation [33] and the analysis method introduced in DWBC [20], we can avoid directly solving this complex functional min-max optimization problem by introducing discriminator-dependent loss correction terms $\mathcal{L}_\pi^{corr}$ and $\mathcal{L}_f^{corr}$ on the losses of policy $\mathcal{L}_\pi$ and dynamics model $\mathcal{L}_f$. In this way, $\pi$, $f$ and $d$ can be efficiently learned by solving three simple minimization problems: $\min_\pi \alpha_\pi \cdot \mathcal{L}_\pi + \mathcal{L}_\pi^{corr}$, $\min_f \alpha_f \cdot \mathcal{L}_f + \mathcal{L}_f^{corr}$ and $\min_d \mathcal{L}_d$, where $\alpha_\pi, \alpha_f \geq 1$ are weight factors for the original losses of $\pi$ and $f$. In the follows, we briefly describe the essential steps of deriving $\mathcal{L}_\pi^{corr}$ and $\mathcal{L}_f^{corr}$, and provide detailed derivations in Appendix A. The outline of DMIL is presented in Appendix B.1.

Denote $x = (s,a,s')$ and $\Omega_{sas'}$ as its domain. Note that the functional $\mathcal{L}_d(d, \log \pi, \log f)$ can be written as the integral of a new functional $F(x, \log \pi, \log f)$ with the following form:

$$\mathcal{L}_d = \int_{\Omega_{sas'}} [P_{D_e}(x) \cdot (-\log d) + P_{D_r}(x) \cdot (-(1 - \log d))]dx \triangleq \int_{\Omega_{sas'}} F(x, d, \log \pi, \log f)dx \tag{4}$$

where we slightly abuse the notations and write the output of $d(s,a,\log \pi(a|s), \log f(s'|s,a))$ as $d$ and $F(x, d, \log \pi, \log f)$ as $F$ hereafter; $P_{D_e}$ and $P_{D_r}$ are distributions of $x$ in $\mathcal{D}_e$ and $\mathcal{D}_r$. To simplify the analysis, we focus on the inner maximization problem in Eq.(3). According to calculus of variation, maximizing $\mathcal{L}_d$ with respect to function $\pi$ and $f$ requires to find the extrema of $\mathcal{L}_d$, which can be achieved by solving the following associate Euler-Lagrangian equations:

$$\begin{cases} F_\pi - \frac{\partial}{\partial x} F_{\frac{\partial \pi}{\partial x}} = F_\pi = 0 \\ F_f - \frac{\partial}{\partial x} F_{\frac{\partial f}{\partial x}} = F_f = 0 \end{cases} \tag{5}$$

where $F_y$ stands for $\frac{\partial F}{\partial y}$. Let $\theta_\pi$ and $\theta_f$ denote the network parameters of policy $\pi$ and dynamics model $f$. Using the analysis on policy $\pi$ as an example. Assuming $F$ and $d$ are continuously differentiable with respect to $d$ and $\log \pi$ respectively, from the first equation in Eq.(5), we have $F_\pi \cdot \frac{\partial \pi}{\partial \theta_\pi} = \frac{\partial F}{\partial d} \cdot \frac{\partial d}{\partial \log \pi} \cdot \frac{\partial \log \pi}{\pi} \cdot \frac{\partial \pi}{\partial \theta_\pi} = \frac{\partial d}{\partial \log \pi} \cdot \frac{\partial F}{\partial d} \cdot \nabla_{\theta_\pi} \log \pi = 0$. As $d$ is determined by the outer minimization problem of Eq.(3), thus $\frac{\partial d}{\partial \log \pi}$ is not obtainable by solely inspecting the inner maximization problem. To ensure the previous equation hold, we can instead consider a relaxed condition by letting $\frac{\partial F}{\partial d} \cdot \nabla_{\theta_\pi} \log \pi = 0$. The integration of this new condition is still 0 ($\int_{\Omega_{sas'}} \frac{\partial F}{\partial d} \cdot \nabla_{\theta_\pi} \log \pi \mathrm{d}x = 0$), which leads to the following tractable condition:

$$- \mathop{\mathbb{E}}_{(s,a,s')\sim\mathcal{D}_e} \left[ -\frac{1}{d} \cdot \nabla_{\theta_\pi} \log \pi \right] + \mathop{\mathbb{E}}_{(s,a,s')\sim\mathcal{D}_r} \left[ -\frac{1}{1-d} \cdot \nabla_{\theta_\pi} \log \pi \right] = 0 \qquad (6)$$

Above can be equivalently perceived as the first-order optimality condition of minimizing the following corrective loss term $\mathcal{L}_\pi^{corr}$ for policy $\pi$:

$$\mathcal{L}_\pi^{corr} = \mathop{\mathbb{E}}_{(s,a,s')\sim\mathcal{D}_e} \left[ \frac{1}{d} \cdot \log \pi(a|s) \right] - \mathop{\mathbb{E}}_{(s,a,s')\sim\mathcal{D}_r} \left[ \frac{1}{1-d} \cdot \log \pi(a|s) \right] \qquad (7)$$

Similarly, we can obtain the corrective loss term $\mathcal{L}_f^{corr}$ for dynamics model $f$ as:

$$\mathcal{L}_f^{corr} = \mathop{\mathbb{E}}_{(s,a,s')\sim\mathcal{D}_e} \left[ \frac{1}{d} \cdot \log f(s'|s,a) \right] - \mathop{\mathbb{E}}_{(s,a,s')\sim\mathcal{D}_r} \left[ \frac{1}{1-d} \cdot \log f(s'|s,a) \right] \qquad (8)$$

### 2.3 Extensions to Scenarios with Additional Suboptimal Dataset

The DMIL framework can be easily extended to IL scenarios with a small expert dataset $\mathcal{D}_e$ and a larger dataset $\mathcal{D}_o$ sampled from one or multiple potentially suboptimal policies [13, 20, 34]. Under this setting, we can add a second optimality discriminator $d_o$ in additional to the original rollout discriminator in DMIL (referred as $d_r$ in this setting), dedicated to differentiate between expert and suboptimal samples in both $\mathcal{D}_o$ and $\mathcal{D}_r$. We follow Xu et al. [20] to adopt a positive-unlabeled (PU) learning [35] objective for $d_o$, and also introduce a second pair of adversarial relationship between $\pi$ and $d_o$. PU-learning enables learning from positive (expert data $\mathcal{D}_e$) and unlabeled data ($\mathcal{D}_o \cup \mathcal{D}_r$ in our case) with a hyperparameter $\eta$ to capture the proportion of positive samples to unlabeled samples.

$$\min_{d_o} \max_{\pi} \mathcal{L}_{d_o} := \eta \mathop{\mathbb{E}}_{(s,a)\sim\mathcal{D}_e} [-\log d_o(s, a, \log \pi(a|s))] +$$
$$\mathop{\mathbb{E}}_{(s,a)\sim\mathcal{D}_o\cup\mathcal{D}_r} [-\log(1 - d_o(s, a, \log \pi(a|s)))] - \eta \mathop{\mathbb{E}}_{(s,a)\sim\mathcal{D}_e} [-\log(1 - d_o(s, a, \log \pi(a|s)))] \qquad (9)$$

Similar to the derivation in previous section, when jointly solving above functional min-max optimization problem together with Eq.(1)-(3), we can obtain the following updated corrective loss term for policy $\pi$, which now depends on outputs of both discriminators $d_o$ and $d_r$, with $\beta_o$ and $\beta_r$ being the weight parameters for the two discriminators. We term this extension as Dual-Discriminator guided Model-based Imitation Learning (D2MIL). Complete derivation can be found in Appendix A.

$$\mathcal{L}_\pi^{corr} = \mathop{\mathbb{E}}_{(s,a,s')\sim\mathcal{D}_e} \left[ \left( \frac{\beta_o \eta}{d_o(1-d_o)} + \frac{\beta_r}{d_r} \right) \cdot \log \pi(a|s) \right] - \mathop{\mathbb{E}}_{(s,a,s')\sim\mathcal{D}_o} \left[ \left( \frac{\beta_o}{1-d_o} - \frac{\beta_r}{d_r} \right) \cdot \log \pi(a|s) \right]$$
$$- \mathop{\mathbb{E}}_{(s,a,s')\sim\mathcal{D}_r} \left[ \left( \frac{\beta_o}{1-d_o} + \frac{\beta_r}{1-d_r} \right) \cdot \log \pi(a|s) \right] \qquad (10)$$

## 3 Experiments

We evaluate our methods against offline IL baseline methods on both D4RL benchmark datasets [29] and a real-world wheel-legged robot. Our methods achieve superior performance and robustness compared with baselines, especially under small datasets. Experiment setups and results are described below. Ablation study on the impact of different design elements of DMIL can be found in Appendix C.2. Implementation details and extra comparative results are reported in Appendix B and C.

Table 1: Normalized scores for models trained on different proportion of D4RL MuJoCo-expert datasets and Adroit-human tasks. Results are averaged over 3 random seeds.

| | Ratio | BC | BC+d | 2-phase BC+d | DWBC+d | ValueDICE | IQ-Learn | DMIL |
|---|---|---|---|---|---|---|---|---|
| Hopper | 100% | 95.06±20.38 | 106.78±4.4 | **110.59±0.63** | 96.96±18.15 | 60.34±10.12 | 25.49±5.34 | **110.22±1.22** |
| | 10% | 83.52±30.58 | 100.59±13.21 | 104.35±9.44 | 91.52±24.81 | 58.77±10.45 | 25.16±6.69 | **111.56±1.51** |
| | 5% | 73.35±37.04 | 94.82±19.72 | 99.66±14.98 | 88.35±28.16 | 44.94±13.71 | 4.58±0.51 | **111.14±1.83** |
| | 2% | 53.54±36.89 | 61.57±30.18 | 88.24±25.63 | 81.70±32.27 | 31.38±12.84 | 3.72±0.56 | **108.51±3.88** |
| Halfcheetah | 100% | 91.95±1.24 | 89.23±1.35 | 91.48±0.33 | 83.75±6.57 | 56.07±5.33 | 38.12±9.96 | **93.34±1.29** |
| | 10% | 90.64±2.21 | 89.71±2.88 | 71.27±19.33 | 77.48±12.97 | 48.77±8.30 | 18.36±16.09 | **92.69±1.82** |
| | 5% | 82.90±11.71 | 76.40±16.94 | 70.89±23.06 | 65.76±20.55 | 30.61±6.98 | 7.12±6.77 | **90.18±4.43** |
| | 2% | 23.58±16.36 | 21.48±16.86 | 57.48±25.63 | 30.10±22.27 | 17.47±7.63 | 1.63±1.37 | **76.87±15.31** |
| Walker2d | 100% | 107.35±2.29 | 106.82±1.33 | **108.15±0.27** | 103.92±6.53 | 86.42±11.20 | 100.96±1.23 | 107.65±0.37 |
| | 10% | 105.36±4.38 | **107.61±1.14** | 106.40±1.96 | 91.17±25.05 | 86.76±13.04 | 73.65±12.64 | 107.62±0.83 |
| | 5% | 103.21±7.81 | 105.42±3.93 | 104.51±4.54 | 89.78±24.81 | 83.51±12.96 | 59.47±23.17 | **107.89±0.71** |
| | 2% | 58.34±35.86 | 60.64±35.10 | 86.71±21.20 | 65.19±36.27 | 78.84±23.16 | 34.19±20.11 | **105.55±4.42** |
| pen-human | | 57.91±55.05 | 7.27±15.87 | **68.57±53.57** | 18.61±26.46 | 52.51±19.58 | 4.94±11.51 | 67.56±57.87 |
| hammer-human | | 1.05±1.01 | 1.18±1.25 | 1.64±1.30 | 0.67±0.64 | 1.12±0.64 | 0.37±0.13 | **2.06±1.91** |
| door-human | | 0.47±0.65 | 0.16±0.29 | 0.94±1.24 | 0.01±0.21 | 0.22±0.01 | -0.28±0.01 | **6.06±7.56** |

## 3.1 Experiment Setup

**Baselines.** We compare DMIL with 5 baselines: 1) BC: vanilla BC [2]; 2) BC+d: learns a dynamics model alongside BC to generate rollouts, and the policy is trained on both expert and rollout data; 3) 2-phase BC+d: first pretrains the dynamics model and a BC policy on expert data, then uses BC+d to fine-tune the policy; 4) DWBC+d: we use a pretrained dynamics model and a BC policy to generate the suboptimal dataset required in DWBC, and then run DWBC to learn the policy; 5) ValueDICE: we implement an offline version of the original ValueDICE [9], which uses a learned dynamics model to serve as the online sampling environment; 6) IQ-Learn [36]: a recent IL method that learns Q function to implicitly represent the policy, and can work offline. For D2MIL, we compare it with BC trained on expert data only (BC-exp) and on all data (BC-all), as well as two recent methods ORIL [13], DemoDICE [34] and DWBC [20] which are designed for the same problem setting.

**Simulation Tasks.** We conduct the experiments on the widely-used D4RL [29] MuJoCo expert/medium datasets and the more complex Adroit human datasets (Pen, Hammer, Door). To investigate the impact of sample size on model performance, we randomly sample certain proportions of transitions from MuJoCo expert datasets to construct a set of much smaller datasets for evaluation.

**Real-world Robotic Tasks.** We also experiment on a real-world robot which stands on a pair of wheels to get balanced, as shown in Figure 3. The states of robot are composed of its forward tilt angle $\theta$, displacement $x$, angular velocity $\dot{\theta}$ and linear velocity $\dot{x}$. The robot is controlled by the torque $\tau$ of motors at two wheels. We evaluate our method on two tasks: (1) **Standing still**: keep the robot balanced and not fall down; (2) **Moving straight**: keep the robot balanced and move forward with a target velocity $v$. The dataset for these tasks are collected from very few human demonstrations (10,000 transitions from about 50s human control at a sampling frequency of 200Hz).

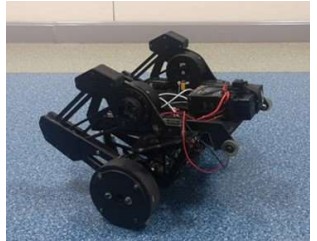

Figure 3: Wheel-legged robot

## 3.2 Results

**Comparative Evaluation on D4RL Benchmarks.** The comparative results are presented in Table 1. We can see that in many tasks, naïvely incorporating dynamics model with BC only leads to marginal improvement. This is due to the lack of discrimination on the quality of rollout data. 2-phase BC+d that use a pretrained, high quality dynamics model and rollout policy in some cases can result in improved performance under small dataset. Besides, offline ValueDICE performs poorly owing to its reliance on accurate online interaction. IQ-Learn performs badly on the continuous control tasks with high-dimensional state-action space. For DWBC+d, we can see that simply incorporating

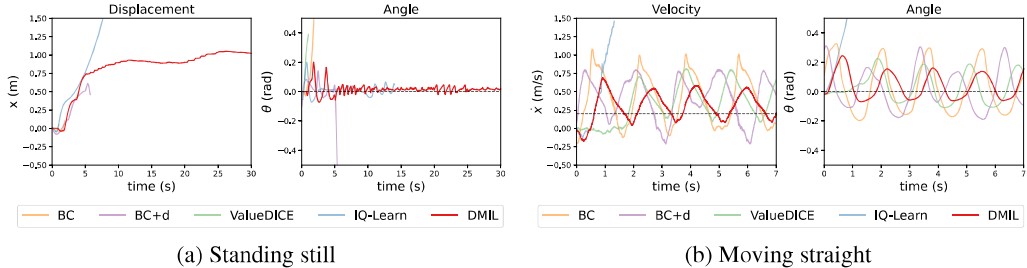

(a) Standing still                    (b) Moving straight

Figure 4: Evaluation results on a real-world wheel-legged robot

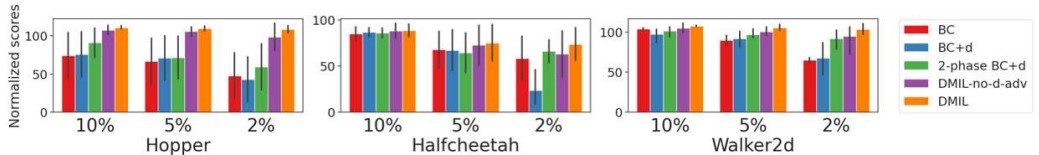

Figure 5: Evaluation results on policy robustness. For different sizes of expert datasets, we randomly pick 20% samples and add a Gaussian noise on the states to make policy learning more challenging.

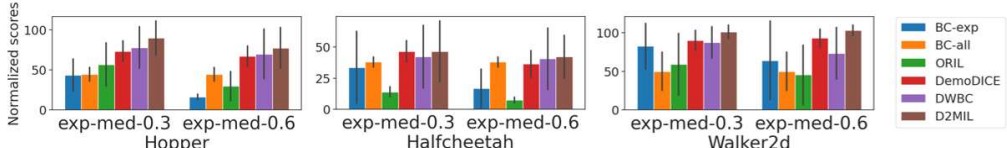

Figure 6: Evaluation results of D2MIL under small datasets. We first sample 1% trajectories from the D4RL MuJoCo expert datasets. We then sample $X$ proportion of these trajectories and combine them with the 2% medium dataset to constitute the suboptimal dataset $\mathcal{D}_o$. The remaining 1-$X$ trajectories constitute the expert dataset $\mathcal{D}_e$. The larger $X$, $\mathcal{D}_o$ contains more high quality data, but corresponds to a smaller expert dataset $\mathcal{D}_e$. We label each task as exp-med-$X$ in the figure.

rollouts from dynamics model as suboptimal dataset in DWBC brings no benefit to dynamics model learning, and insufficient leverage of information in the limited data, which leads to substaintial performance drop with smaller dataset. By contrast, our method achieves the best performance in almost all tasks with small variance. Most importantly, we find DMIL performs surprisingly well under small datasets while other baselines suffer from severe performance degeneration. It achieves comparable performance even if the training data is reduced to 5% or 2% of its original size.

**Comparative Evaluation on Real-World Tasks.**  The imitation performance of two tasks on a wheel-legged robot are shown in figure 4. In these two tasks, we only use 50s human demonstrations to learn the policy. For the Standing still task, despite some small drifts, the robot using DMIL policy can maintain in a balanced state for over 30s, which achieves the most stable performance in both displacement measure and tilt angle. The robot with other control policies either quickly bump to the ground (BC,BC+d,valueDICE) or dashes forward (IQ-learn). For the Moving straight task, most methods can make the robot move forward within a certain speed range, but DMIL policy maintains a closest speed to the target speed $v$=0.2m/s and also keeps a relatively more balanced state.

**Evaluation on Policy Robustness.**  We further evaluate the policy robustness of DMIL under small and noisy training data on MuJoCo tasks in Figure 5. We compare with three stronger baselines in Table 1: BC, BC+d and 2-phase BC+d. To further examine the effectiveness of the cooperative-yet-adversarial learning scheme on the learned dynamics model, we add an additional baseline DMIL-no-d-adv, which removes $\mathcal{L}_f^{corr}$ as well as $\log f$ in the input of discriminator $d$ from DMIL. We observe that the performances of BC and BC+d drop with the introduction of noise, mainly due to the lack of discrimination on data quality. 2-phase BC+d is slightly better, but still perform worse than DMIL and DMIL-no-d-adv. Due to the absence of adversarial learning in dynamics model, DMIL-no-d-adv is generally less performant compared with DMIL due to the noisy training data. In all tasks, DMIL shows great robustness to training noise and achieves almost the same performance

as the case without noise (Table 1). This is because that the discriminator of DMIL in this setting not only distinguishes dynamics correctness and optimality of rollouts, but can also serve as a denoiser to identify and alleviate the negative impact of noisy inputs for policy and dynamics model.

**Evaluation of D2MIL.** We also evaluate the performance of D2MIL when learning with a small expert dataset and a larger suboptimal dataset in Figure 6. The results show that D2MIL outperforms state-of-the-art method DWBC [20] and other baselines in all tasks. The introduction of the dynamics model $f$ and the two discriminators ($d_r$ and $d_o$) indeed help with improving the generalization performance of imitating policy under small datasets, which demonstrates the effectiveness of D2MIL in scenarios with suboptimal data.

## 4 Related Work

**Model-based Imitation Learning.** To combat the covariate shift and improve sample efficiency, many online IL studies have incorporated dynamics models during policy learning [27, 37, 38, 39]. These methods typically require online system interactions or additional expert guidance to correct model errors. Under offline settings without environment interaction, incorporating the model-based approach is much more challenging and less explored. A few existing works all bear some limitations, such as requiring an extra suboptimal dataset [26] or a misspecified simulator [28], only applicable to imagery input [40], or simply fully trust the learned model [27]. Many of these methods assume sufficient coverage of demonstration data, which can be fragile in scenarios with small datasets.

**Offline Imitation Learning.** Offline IL methods that imitate expert demonstrations can be categorized into two paradigms, behavior cloning (BC) and offline inverse reinforcement learning (offline IRL). BC [2] is the simplest IL method, it trains a policy by maximizing the log-likelihood of observed actions. Some recent works enhance BC by using energy-based model [41, 42] or introducing curriculum training strategy [43]. Offline IRL methods [9, 44, 45, 42, 36] on the other hand, consider matching the reward or state-action distribution of the expert policy. This can be done explicitly by learning a reward function [44] or implicitly by learning a Q-function that represents both reward and policy [9, 36]. Although these recent methods can mitigate covariate shift to some extent, they still struggle to work under limited expert data and suffer from the involvement of suboptimal data.

Another stream of studies focus on the problem when demonstrations contain suboptimal data. Some studies [46, 47] leverage previously learned policies [47] or entropy of the model [46] as weights to penalize noisy demonstrations. However, they require the clean expert data occupy the majority of the offline dataset. When both the expert demonstrations and additional suboptimal data are given, some IRL-based methods [13, 26, 34] first construct a reward function to distinguish expert and suboptimal data, and then use it to solve an offline RL problems. The drawbacks of these methods are that the reward learning through offline IRL is costly, and the inner-loop offline RL problem also suffers from training instability [19]. The recently proposed DWBC [20] trains a discriminator to distinguish expert and non-expert data and uses its outputs to re-weight the IL objective, so as to imitate demonstrations selectively. Our method shares some similarity with DWBC, however, we use the discriminator to distinguish both the dynamics discrepancy and suboptimality of model rollout data, and re-weight the objectives of both the IL policy and the dynamics model.

## 5 Conclusion and Limitations

We propose a model-based offline IL framework DMIL for scenarios with limited expert data, which is composed of an imitation policy, a dynamics model and a discriminator. We use the discriminator as a bridge to couple the learning process of all three models through a cooperative-yet-adversarial learning scheme. This design allows us fully leverage the generalizability of dynamics model to improve state-action space coverage, while also alleviating the negative impacts from potentially problematic rollouts. Our framework can also be extended to scenarios with suboptimal data (D2MIL). Through comprehensive experiments, we show that our method achieves strong performance and robustness under small datasets, which can be a nice tool for many real-world IL tasks.

Our method also has some limitations. When the state-action space is large or the MDP is partially observed, the dynamics model might need to be specially designed. For future directions, adopting temporal models, or learning the dynamics in latent state space might be a solution to achieve improved model performance.

## Acknowledgments

We thank the anonymous reviewers for their thoughtful feedback. This work is supported by funding from Haomo.AI.

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
