# OpenReview forum: "Discriminator-Guided Model-Based Offline Imitation Learning"
_robot-learning.org/CoRL/2022/Conference — CoRL 2022 Poster_

### Official Review · Reviewer_8kmH · 2022-07-31

**Originality:** Good
**Technical Quality:** Good
**Clarity Of Presentation:** Very Good
**Impact:** 3

**Recommendation:**

Weak Reject: I recommend rejecting the paper, but will not argue for my recommendation if the majority of other reviewers have a different opinion.

**Summary:**

This paper presents a model-based offline imitation learning method that trains a model, a policy and a discriminator in both a cooperative and adversarial way to improve the robustness of the dynamics model and the the performance of the policy. The method is evaluated in simulated domains with both expert demonstrations and sub-optimal offline data settings and is shown to be superior compared to prior offline imitation learning methods. The authors also include the evaluation on the real-world wheel-legged robot where the method appears to be effective.

**Issues:**

1. Compare to model-based baselines such as [35, 36] and also DemoDICE [45].

2. Include comparisons to [19] in all domains.

3. Clarify hyperparameter selection schemes.

**Quality Of The Limitations Section:**

Limitations are addressed clearly

**Reviewer Expertise:**

4: The reviewer is confident but not absolutely certain that the evaluation is correct

**Robotics Focus:**

Sufficient demonstration on hardware

**Strengths And Weaknesses:**

**Strengths:**

1. The paper presents a new model-based offline method that trains a discriminator along with the learned policy and dynamics model, unlike prior model-based IL methods. The discriminator takes the dynamics model as an additional input, which is different from the prior discriminator-based IL works [19]. Such a scheme seems to effectively achieve a cooperative yet adversarial game among the discriminator, dynamics model and the policy. Moreover, the method achieves strong results in the simulated D4RL domains with fully expert or human data.

2. The authors include ablations that show the importance of $L_corr$ and the log-likelihood of the dynamics model as inputs to the discriminator, showing the effectiveness of the algorithmic contributions in practice.

**Weakness**:

1. Despite that the method gives the cooperative yet adversarial game, the technical part of the method appears to be a model-based version of the prior work [19], which is rather straightforward and makes the novelty a bit limited. The derivation of the algorithm is also heavily based on [19].

2. I'm not sure why comparisons to [19] are not included in Table 1 and Figure 19 as the difference between the proposed method and [19] is rather small in Figure 5 and 6. It is important to compare to [19] everywhere as it appears to be the strongest baseline. Moreover, I think it is important to compare to some model-based IL methods such as [35, 36]. While these methods are proposed to be online, they should be easy to be applied in the fully offline setting. I also think DemoDICE [45] should be a good baseline to include as it handles offline IL with suboptimal data.

3. Finally, I wonder how the hyperparameters are tuned. The method introduces lots of hyperparameters and it's known that tuning hyperparameters in offline IL/RL usually requires online rollouts, which kind of defeats the purpose of fully offline training particularly in the case where there are so many hyperparameters to tune.

**Summary Of Recommendation:**

Based on the Strengths And Weaknesses section, I'm leaning towards a weak reject of the paper but I can change my rating if all the concerns are properly addressed.

---

> ### Author Response · Authors · 2022-08-25
> **Response to Reviewer 8kmH**
>
> We sincerely appreciate the reviewer for the thoughtful and detailed comments, especially the positive feedback on our work. Regarding the concerns from the reviewer, we describe the discussion as follows:
>
>
> >**Q1. Compare to model-based baselines such as [35, 36] and also DemoDICE [45].**
>
> - VMAIL [35] is an online IL methods specially designed for visual inputs problem. It projects the visual inputs into latent space and then trains a dynamics model in it. Our method is a more generic and domain-independent offline IL algorithm, which does not rely on latent space dynamics model, thus the two methods are not directly comparable. Nevertheless, we have found the source code of VMAIL and tried to modify it into an offline version with general inputs, but the official code does not seem to work. For reference [36], it is a relatively outdated online IL method published in 2013 with no source code. This method heavily rely on online data collection for probabilistic model improvement, hence not very compatible with our offline IL setting.
> - For DemoDICE [45], we have added it as an additional baseline for D2MIL in our revised paper. The evaluation results are updated in Figure 6 and also presented in the table below. For brevity, we term the setting 'exp-med-0.3' as '-0.3', and 'exp-med-0.6' as '-0.6'. DemoDICE has comparable performance as strong baselines such as DWBC, but still underperforms D2MIL in most tasks.
>
> |task|DemoDICE|DWBC|D2MIL
> |---|---|---|---|
> |Hopper-0.3|72.49 $\pm$ 13.65|78.32 $\pm$ 26.85|**89.83 $\pm$ 21.12**
> |Hopper-0.6|67.55 $\pm$ 14.68|70.62 $\pm$ 31.32|**77.47 $\pm$ 25.88**
> |Halfcheetah-0.3|**46.81 $\pm$ 9.02**|41.62 $\pm$ 24.66|**46.45 $\pm$ 24.40**
> |Halfcheetah-0.6|36.40 $\pm$ 11.47|41.56 $\pm$ 24.03|**42.39 $\pm$ 17.07**
> |Walker2d-0.3|91.39 $\pm$ 13.52|88.11 $\pm$ 19.69|**101.03 $\pm$ 8.95**
> |Walker2d-0.6|94.14 $\pm$ 12.44|74.35 $\pm$ 33.95|**103.16 $\pm$ 6.77**
>
>
> ---
>
> >**Q2. Include comparisons to [19] in all domains.**
>
> - We have discussed the distinctions of our method and DWBC in the public response. We also add an additional baseline DWBC+d for the evaluation of DMIL. Please refer to "Response to All" and our revised paper for more details.
>
> ---
> >**Q3. Clarify hyperparameter selection schemes.**
>
> - Thank you for the comment. Please refer to "Response to All" as well as Appendix B.2 in our paper for detailed discussion and information.

---

### Official Review · Reviewer_wiFv · 2022-07-31

**Originality:** Very Good
**Technical Quality:** Excellent
**Clarity Of Presentation:** Excellent
**Impact:** 4

**Recommendation:**

Weak Accept: I recommend accepting the paper, but will not argue for my recommendation if the majority of other reviewers have a different opinion.

**Summary:**

The paper presents a new approach for offline imitation learning. The key idea is to learn a dynamics model along with the policy. To do so they use a discriminator that takes into account both dynamics and policy simultaneously. They show that this approach improves performance of agents even when the number of expert demonstrations available is low. They also show on a real robot their methods works better than other baselines.

**Issues:**

1. Adding ablation experiments with different loss terms and sensitivity of their method to different hyper-parameter settings.


**Quality Of The Limitations Section:**

Limitations are addressed clearly

**Reviewer Expertise:**

4: The reviewer is confident but not absolutely certain that the evaluation is correct

**Robotics Focus:**

Sufficient demonstration on hardware

**Strengths And Weaknesses:**

Strengths
1) The proposed approach has solid performance over different expert dataset sizes.
2) The proposed method also outperforms many different offline imitation learning algorithms.
3) One of the baselines proposed by the authors "2 phase BC+d" is a very reasonable baseline that outperforms more complicated algorithms. More details on this baseline would be useful.
4) Real robot experiments showing the effectiveness of their approach.

Weakness
1) Ablation experiments missing the show the importance of each term in the loss function. This will give more insight into why their method is performing so well. It seems only in Fig. 5 when the authors test the robustness of policy with noisy datasets, they have an ablation experiment.

Minor Weakness
1) "performance degeneration under limited expert data due to covariate shift" - this statement in the abstract is not backed up by any experiment. Performance degeneration is usually due to drift from expert demonstrations' data distribution rather than covariate shift.



**Summary Of Recommendation:**

The proposed approach performs well when the number of demonstrations available are low and there is noise present in the data. The authors have made extensive comparison with existing methods. They have also shown their method works on a real robot.

---

> ### Author Response · Authors · 2022-08-25
> **Response to Reviewer wiFv**
>
> We would like to thank the reviewer for the thoughtful comments and suggestions. Regarding the concerns from the reviewer, we describe the detailed response as follows:
>
>
> >**Q1. Ablation experiments missing the show the importance of each term in the loss function. This will give more insight into why their method is performing so well. It seems only in Fig. 5 when the authors test the robustness of policy with noisy datasets, they have an ablation experiment.**
>
> - We have provide a comprehensive ablation study in Section C.2 of the Appendix (included in our supplementary material), which includes comparision with following baselines:
>   - **DMIL-no-d-adv**: removing the coupling and the adversarial relationship between discriminator $d$ and dynamics model $f$.
>   - **DMIL-no-d-adv\&$\pi$-info**: on the basis of DMIL-no-d-adv, we further remove the additional information $\log \pi$ from the inputs of $d$.
>   - **2-phase BC+d**: this baseline can be perceived as the reduction of DMIL that completely removes the cooperative-yet-adversarial learning scheme.
>   - **BC** and **BC+d**: minimal baselines without or with a dynamics model used for comparison.
> - Detailed results are presented in Table 5 of Appendix C.2. From the results, we can see that the cooperative-yet-adversarial learning scheme involving $\pi$, $f$ and $d$ indeed help with improving policy robustness and imitation performance. We also add a sentence in Section 3 in the revised paper to indicate the additional ablation results in the Appendix.
>
> ---
> >**Q2. Minor Weakness: "performance degeneration under limited expert data due to covariate shift" - this statement in the abstract is not backed up by any experiment. Performance degeneration is usually due to drift from expert demonstrations' data distribution rather than covariate shift.**
>
> - Thank you for the suggestion. The "covariate shift" in most papers is defined as the case of distinct training and test distributions in a learning problem (reference [1-5] listed below). It is a specific type of dataset shift. In our paper, our intention is to show that when the expert dataset is rather small, it is difficult to cover the entire expert data distribution, which in turn leads to covariate shift problem. We have modified the abstract to make the descriptions more accurate and avoid misunderstanding.
>
> ---
> >**Issues: 1. Adding ablation experiments with different loss terms and sensitivity of their method to different hyper-parameter settings.**
>
> - Detailed ablation experiments are included in Appendix C.2. And for hyperparameter settings, we have discussed in Appendix C.3, please also refer to the discussion in 'Response to All'.
>
> ---
> **References:**
>
> [1] N. Rajaraman, L. F. Yang, J. Jiao, and K. Ramchandran. Toward the fundamental limits of imitation learning. In Proceedings of the 34th International Conference on Neural Information Processing Systems, NIPS’20, 2020.
>
> [2] J. C. Spencer, S. Choudhury, A. Venkatraman, B. D. Ziebart, and J. A. Bagnell. Feedback in imitation learning: The three regimes of covariate shift. CoRR, abs/2102.02872, 2021.
>
> [3] G. Tennenholtz, A. Hallak, G. Dalal, S. Mannor, G. Chechik, and U. Shalit. On covariate shift of latent confounders in imitation and reinforcement learning. In International Conference on Learning Representations, 2022.
>
> [4] G. D. Y, N. G. Nair, P. Satpathy and J. Christopher. Covariate Shift: A Review and Analysis on Classifiers. 2019 Global Conference for Advancement in Technology (GCAT), 2019.
>
> [5] Bickel S, Brückner M, Scheffer T. Discriminative learning under covariate shift. Journal of Machine Learning Research, 2009, 10(9).

---

### Official Review · Reviewer_F1vv · 2022-07-31

**Originality:** Good
**Technical Quality:** Good
**Clarity Of Presentation:** Good
**Impact:** 3

**Recommendation:**

Weak Accept: I recommend accepting the paper, but will not argue for my recommendation if the majority of other reviewers have a different opinion.

**Summary:**

The paper introduces an approach to offline imitation learning that extends a dynamic forward model with an adversarial discriminator. The idea is to use the discriminator to estimate the forward model correctness and optimality of the model rollouts at the same time. Additionally, the authors extend the method to incorporate sub-optimal data as well.

**Issues:**

* Clarity of presentations. See the comments above.

**Quality Of The Limitations Section:**

Additional details required

**Reviewer Expertise:**

4: The reviewer is confident but not absolutely certain that the evaluation is correct

**Robotics Focus:**

Sufficient demonstration on hardware

**Strengths And Weaknesses:**

# Strengths
* The method is novel. I like the idea of re-evaluating the correctness of the predictions of the forward model using a discriminator. The existing techniques use online interactions (see [1, 2]) instead of a model that is accurate and does not require re-evaluation. Or they use ensembles of forward models, which are computationally expensive (PETS, [3]). Using a discriminator mitigates the issues associated with computational costs of PETS.
* The method is theoretically sound and outperforms on the baselines. In particularly, the method outperforms the baselines in the low-data regime which is considered to be the most challenging for offline learning.

# Weaknesses
* The clarity of writing can be improved. It is not clear how the method is implemented. The paper can be improved by adding a pseudo-code for the algorithm.

[1] Generative adversarial imitation learning, J. Ho, S. Ermon

[2] Discriminator-Actor-Critic: Addressing Sample Inefficiency and Reward Bias in Adversarial Imitation Learning, I. Kostrikov, K. Agrawal, D. Dwibedi, S. Levine, J. Tompson

[3] Deep reinforcement learning in a handful of trials using probabilistic dynamics models, K. Chua, R. Calandra, R. McAllister, S. Levine

**Summary Of Recommendation:**

In general, I recommend this paper for acceptance. However, the clarity of presentation can be improved. See my comments above.

---

> ### Author Response · Authors · 2022-08-25
> **Response to Reviewer F1vv**
>
> We thank the reviewer for the comments and positive feedback. Regarding the question from the reviewer, we describe the response as follows:
>
> >**Q1. The clarity of writing can be improved. It is not clear how the method is implemented. The paper can be improved by adding a pseudo-code for the algorithm.**
>
> - We have moved the pseudo-code of DMIL from Appendix to Section 2.2, as well as elaborate more details in the main text of our revised paper. The complete derivation and implementation details are included in Appendix A and B due to the page limit. Please refer to the Appendix of our paper for additional discussion and details of our approach.

---

### Official Review · Reviewer_ihkf · 2022-08-01

**Originality:** Good
**Technical Quality:** Good
**Clarity Of Presentation:** Good
**Impact:** 3

**Recommendation:**

Weak Reject: I recommend rejecting the paper, but will not argue for my recommendation if the majority of other reviewers have a different opinion.

**Summary:**

The paper presents a model-based offline imitation leaning algorithm – DMIL. The method jointly trains a policy, a dynamics model, and a discriminator in an iterative algorithm. The policy and the dynamics model are trained with maximum likelihood on the expert data, with further regularization (corrective loss terms) using the discriminator. The discriminator is trained with a GAN-style objective on the expert and rolled-out data. Crucially, the discriminator includes the log-likelihood of the policy and the dynamics model as inputs to facilitate information sharing. An extension (D2MIL) is also presented for when sub-optimal data is available. Experiments on D4RL MuJoCo and Adroit tasks, and a real-world wheeled robotic task show that DMIL compare favorably to the baseline offline imitation methods especially when expert data is limited.

**Issues:**

1. Differentiation from prior work: The “cooperative information sharing” for the discriminator was introduced in DWBC (Xu et al.), where they used $\log\pi$ in the discriminator input. DMIL builds upon this idea by introducing a learned dynamics model and augmenting the discriminator input with the model likelihood. The corrective loss terms follow the same derivation as DWBC. Although I believe that the extensions are important, the current presentation in Section 2.2 could be improved to cover DWBC in a bit more detail and clearly outline the new contributions of this paper.


2. Other suggestions regarding presentation:

* &nbsp; It would really help to move the DMIL algorithm box (Appendix B.1) to the main paper. The algorithm box provides a full picture of the approach that is currently difficult to glean from Section 2.

* &nbsp; I would also recommend that you move the no-$\log f$-no-$\log\pi$ ablation results from the Appendix (Table 5) to the main paper results, and comment on the performance comparison to DMIL. This seems to be an important ablation to justify the “cooperative information sharing” argument.

* &nbsp; Section 2.3: some details could be included here to make the paper self-contained. For instance, the acronym “PU” is not expanded, $\eta$ is not defined, etc.

3. MuJoCo environments – it’s nice to see that for Hopper/Walker/Cheetah, using only about 5% of the expert data recovers the performance of using the full data. However, I wonder if this is largely due to the nature of the environments – they are deterministic and involve repetitive (cyclic) motions, so only a few expert transitions are usually sufficient. On the more complicated tasks, it seems there’s still big gap to the expert’s performance (especially hammer/door). Could the authors give some intuition/ideas for this?

4. Clarifying question: there’s a relaxation that is made in the derivation of equation 6, where $\frac{\partial d}{\partial \log\pi}$ is ignored. In an iterative algorithm, can this term be computed for the current discriminator using the auto-differentiation tools and used for the policy/dynamics gradient?

**Quality Of The Limitations Section:**

Limitations are addressed clearly

**Reviewer Expertise:**

4: The reviewer is confident but not absolutely certain that the evaluation is correct

**Robotics Focus:**

Relevant but unlikely to deploy to hardware in near future

**Strengths And Weaknesses:**

Improving the sample-efficiency (in terms of expert data size) of offline imitation learning algorithms is important for their practical application. The paper provides an interesting approach to achieve this and backs it up with nice experiments -- for instance, Table 1 shows that DMIL can achieve competitive performance with only 2% of the original data in Hopper and Walker. Another strong point is the inclusion of results on real robot tasks. I feel that the paper has some shortcomings in the current form – presentation of material and clear differentiation from prior work. These are expanded further below.

**Summary Of Recommendation:**

My pre-rebuttal rating is a weak-reject since I have some reservations about the presentation, comparison to prior work, and experiments (see issues below). The paper has several nice elements to it and I would be happy to reconsider after discussion with the authors.

---

> ### Author Response · Authors · 2022-08-25
> **Response to Reviewer ihkf (2/2)**
>
> ---
> >**Q3. MuJoCo environments – it’s nice to see that for Hopper/Walker/Cheetah, using only about 5% of the expert data recovers the performance of using the full data. However, I wonder if this is largely due to the nature of the environments – they are deterministic and involve repetitive (cyclic) motions, so only a few expert transitions are usually sufficient. On the more complicated tasks, it seems there’s still big gap to the expert’s performance (especially hammer/door). Could the authors give some intuition/ideas for this?**
>
> - Thank you for the comment. The Adroit-human tasks (pen/hammer/door) are actually considerably more complex than MuJoCo tasks, due to their high dimension of state and action space, human-induced non-Markovian property, as well as the already very small dataset. The Adroit human datasets in D4RL only include 25 human demonstrations provided in the DAPG repository (detailed dataset statistics see Table 3 in the Appendix). The dataset is already extremely small given the complexity and high-dimensionality of the tasks as compared to MuJoCo tasks. For comparison, the full MuJoCo expert dataset contains 1 million transitions, while the largest dataset in the Adroit-human tasks, Hammer-human, only contains 11,310 transitions. Hence the extremely small dataset and the human-induced non-Markovian properties are the main reasons for the gap with expert's performance in these tasks. However, even in such difficult cases, our method can still achieve better performance than other methods, which further validates the effectiveness of our proposed method.
> - As for Mujoco tasks, although repetitive motions might be involved in the datasets, when the sample size is reduced to 2%, it is still difficult to cover the full state distribution. This can be reflected in the dramatic performance drop of BC and other offline IL methods under the 2\% dataset settings. While for DMIL, the dynamics model and the proposed cooperative-yet-adversarial learning scheme greatly improves the model generalization capability and robustness, which lead to limited performance drop under such extreme cases.
>
> ---
> >**Q4. Clarifying question: there’s a relaxation that is made in the derivation of equation 6, where $\frac{\partial d}{\partial \log\pi}$ is ignored. In an iterative algorithm, can this term be computed for the current discriminator using the auto-differentiation tools and used for the policy/dynamics gradient?**
>
> - For the functional min-max problem in Eq.(14), $\pi$ is determined by inner maximization problem, while $d$ is determined by outer minimization problem. Hence $d$ is under-determined when we try to derive a optimality condition by only inspecting the inner maximization problem.
> - Even if we can use auto-differentiation tools to compute $\frac{\partial d}{\partial \log\pi}$, it should be noted that the $d$ used to compute the derivative during training in an iterative algorithm is not necessarily the solution of the outer minimization problem (i.e. $d=\arg\min_d \mathcal{L}_d$), which can potentially misguide the inner maximization problem. Furthermore, the auto-differentiation result of $\frac{\partial d}{\partial \log\pi}$ does not admit an analytical form where we can perform further integration to obtain a tractable condition like Eq.(19) in Appendix A.1.
> - Therefore, we consider an alternative solution, by setting $\frac{\partial F}{\partial d}\cdot\nabla_{\theta_{\pi}}\log\pi=0$. This is a even stronger condition for problem Eq.(14), in which no matter what $d$ is, we can always satisfy the optimality condition for the extrema, and this also admits a tractable final form after integration.

---

> ### Author Response · Authors · 2022-08-25
> **Response to Reviewer ihkf (1/2)**
>
> We thank the reviewer for the detailed comments. Regarding the concerns from the reviewer, we describe the discussion as follows:
>
> > **Q1. Differentiation from prior work: The “cooperative information sharing” for the discriminator was introduced in DWBC (Xu et al.), where they used $\log\pi$ in the discriminator input. DMIL builds upon this idea by introducing a learned dynamics model and augmenting the discriminator input with the model likelihood. The corrective loss terms follow the same derivation as DWBC. Although I believe that the extensions are important, the current presentation in Section 2.2 could be improved to cover DWBC in a bit more detail and clearly outline the new contributions of this paper.**
>
> - We have discussed the distinctions of our method and DWBC in the public response, please refer to 'Response to All'.
>
> ---
> > **Q2. Other suggestions regarding presentation.**
>
> - Thank you for your valuable suggestions. We have reorganized the structure of paper in the resubmitted version, the detailed modifications are as follows:
>     - The algorithm box of DMIL (Algorithm 1) has been moved to Section 2.2 to add more clarity.
>     - We add more details of D2MIL in Section 2.3, which provides necessary information on PU-learning and corresponding hyperparameter $\eta$.
>     - We still put the results of no-$\log f$-no-$\log\pi$ in Table 5 rather than Table 1, as we have included an additional baseline DWBC+d into the Table 1 as suggested by other reviewers, and the table will be too wide to fit into the page if we include an another column. We think keeping the results of no-$\log f$-no-$\log\pi$ in Table 5 could be more beneficial, as the readers can fully compare DMIL with all its variants to get a more sensible understanding of the contribution of each design element in the model. We have also mentioned in the Section 3 about the additional ablation results in the Appendix, please refer to our revised paper.

---

### Author Response · Authors · 2022-08-25
**Response to All**

We notice that there are some common concerns from reviewers regarding the distinctions of DWBC [19] (reference [20] in our revised paper) and our work, as well as the hyperparameter tuning scheme used in our work. We provide the detailed response as follows:

---

> ### Author Response · Authors · 2022-08-25
> **Response to All (2/2)**
>
> ---
> > **2. Hyperparameter tuning scheme used in this paper.**
> - The hyperparameter tuning effort in our proposed methods is actually very minor. We use the same set of hyperparameters in most of our experiments without tuning.
> - In the proposed DMIL, the hyperparameters involved are $\alpha_\pi$ and $\alpha_f$, which are used to balance the impact of correction loss terms. In our implementation, we simply choose $\alpha_\pi=\alpha_f>1$. In all our experiments, the values of $\alpha_\pi$ and $\alpha_f$ are set to be 10 without tuning (see Tabel 2 in the Appendix), as we find this choice already produces good model performance. To further verify their impact, we conducted additional experiments on Hopper tasks with 2% expert data by setting $\alpha_\pi$ and $\alpha_f$ to different values, the results are presented below. It is found that these hyperparameters generally do not need careful tuning and produce similar performance.
>
> |$\alpha_\pi$, $\alpha_f$|5| 10| 20
> |---|---|---|---|
> |Results|106.07 $\pm$ 7.86|108.51 $\pm$ 3.88|105.79 $\pm$ 8.61
>
>
> - For the extended D2MIL, although we have hyperparemeters $\alpha_\pi$, $\alpha_f$, $\eta$, $\beta_o$ and $\beta_r$ in the model, most of them do not need to be tuned. As in DMIL, we set $\alpha_\pi=\alpha_f=10$. We adopt $\eta=0.5$ as a constant, which is same as in ORIL (reference [13] in the revised paper) and DWBC. In our implementation, we make $\beta_o+\beta_r=1$ to reduce the parameter numbers. $\beta_o$ and $\beta_r$ reflect the trade-off between the reliability and optimality of samples in the suboptimal dataset $\mathcal{D}_o$ and rollout data $\mathcal{D}_r$. When $\beta_o=\beta_r$, D2MIL tends to learn policy with high $d_o$ and $d_r$ samples with similar preference. However, if the suboptimal dataset $\mathcal{D}_o$ is known to have high quality, one can use a larger $\beta_r$ to pay more attention to the quality of rollout data. In such cases, both $d_o$ and $d_r$ will output values close to 1 on $\mathcal{D}_o$ samples, resulting high weights to encourage policy learning on these samples. Conversely, if the expert demonstrations $\mathcal{D}_e$ and suboptimal dataset $\mathcal{D}_o$ has considerably large gap, a large $\beta_o$ should be used to ensure policy learning focus more on those expert-like samples. In practical scenarios, we suggest the practitioners just set $\beta_r=\beta_o=0.5$, which generally leads to reasonably good performance. In our real-world experiments, due to the large quality gap between the expert dataset and suboptimal human demonstrations, we set $\beta_o$ to be slightly larger value ($\beta_o=0.6$, $\beta_r=1-\beta_o=0.4$).
> - We have revised our paper to include the new results of different $\alpha_\pi$ and $\alpha_f$ values. We provide the detailed discussion about our hyperperameter selection scheme and default values in Appendix B.2. The impact of different $\beta_o$ and $\beta_r$ values are also discussed in the last paragraph of Appendix A.2.

---

> ### Author Response · Authors · 2022-08-25
> **Response to All (1/2)**
>
> ---
> > **1. Distinctions between our work and DWBC.**
> - The problem settings and design logics of DWBC and our work are very different. DWBC learns an IL policy given a small expert dataset together with a large potentially suboptimal dataset. While DMIL considers a substantially more challenging setting, with only limited expert dataset is provided. In DWBC, one only needs to handle different optimality levels of real samples. But in our setting, the limited data coverage will pose severe generalization issues on the learned IL policy, hence we introduce a model-based framework to improve the sample efficiency, however, this further requires an algorithm to properly handle the dynamics discrepancy and the suboptimality of the model rollout data. For the setting of DWBC, we also extend our method to D2MIL in Section 2.3, and present simulation and real-world comparative results in Figure 6 and Figure 8 (in Appendix).
> - DWBC involves a discriminator to distinguish the suboptimality of demonstrations in different datasets, and conducts adversarial learning between the policy and discriminator. While in DMIL, we construct a "three-party game" among the policy, dynamics model and the discriminator.
> In DMIL, the policy and dynamics model are jointly adversarial to the discriminator. The discriminator not only distinguishes the suboptimality of data samples, but also the dynamics discrepancy. Moreover, the discriminator serves as a bridge to utilize coupled information shared from the policy and dynamics model, and further guide their training processes. This is a very different design as compared to DWBC. Note that simply incorporate the rollout data from dynamics model as the suboptimal dataset in DWBC, without such a "three-party game" design, will not establish effective coupling among the policy, dynamics model and discriminator, which brings no benefit to dynamics model learning and insufficient leverage of the information in the limited data. To demonstrate this, we add an additional baseline DWBC+d in Table 1 of the revised paper (results also presented in the table below). The DWBC+d baseline uses rollout data from a fully trained dynamics model and a BC policy to generate the suboptimal dataset required in DWBC, and then run DWBC to learn the policy. It can be observed that DWBC+d suffers substaintial performance drop with smaller dataset, while DMIL is less impacted.
> - Besides, DWBC adopts a positive-unlabeled (PU) learning objective for the discriminator with additional hyperparameter $\eta$ to model the proportion of expert data in the suboptimal dataset. DMIL directly uses the cross entropy loss to distinguish the real and model rollout data, which is more suitable for the model-based setting.
>
> |task|ratio|DWBC+d|DMIL
> |---|---|---|---|
> |Hopper|100\%|96.96$\pm$18.15|110.22$\pm$1.22
> |Hopper|10\%|91.52$\pm$24.81|111.56$\pm$1.51
> |Hopper|5\%|88.35$\pm$28.16|111.14$\pm$1.83
> |Hopper|2\%|81.70$\pm$32.27|108.51$\pm$3.88
> |Halfcheetah|100\%|83.75$\pm$6.57|93.34$\pm$1.29
> |Halfcheetah|10\%|77.48$\pm$12.97|92.69$\pm$1.82
> |Halfcheetah|5\%|65.76$\pm$20.55|90.18$\pm$4.43
> |Halfcheetah|2\%|30.10$\pm$22.27|76.87$\pm$15.31
> |Walker2d|100\%|103.92$\pm$6.53|107.65$\pm$0.37
> |Walker2d|10\%|91.17$\pm$25.05|107.62$\pm$0.83
> |Walker2d|5\%|89.78$\pm$24.81|107.89$\pm$0.71
> |Walker2d|2\%|65.19$\pm$36.27|105.55$\pm$4.42
> |pen-human|100\%|18.61$\pm$26.46|67.56$\pm$57.87
> |hammer-human|10\%|0.67$\pm$0.64|2.06$\pm$1.91
> |door-human|5\%|0.01$\pm$0.21|6.06$\pm$7.56

---

### Author Response · Authors · 2022-08-25
**Revision Summary and the Revised Version of Paper**

We would like to thank the reviewers and AC for their detailed review comments and the concrete suggestions. We respond to the individual reviews below. We’ve also updated the paper with a number of modifications to address the suggestions and concerns from the reviewers. Summary of changes in the updated version of the paper are as follows:
1. We include more details of our methods in the main text and move the algorithm box of DMIL from Appendix B.1 to Section 2.2 to make the presentation of the paper more clear.
2. We included additional baseline DWBC+d, which is an extension of DWBC with the rollout data from dynamics model as suboptimal data. We report its performance in Table 1 and comment upon the differences of DWBC against our method in the Experiments section.
3. We add comparative experiments on DemoDICE and report the results in Figure 6.
4. We add additional results as well as a new section in the Appendix C.3 to demonstrate the robustness of DMIL under different choices of hyperparameters.
5. We have added several additional references mentioned by the reviewer.

---

### Meta-Review · Area_Chair_Fv1e · 2022-08-13

**Recommendation:** Accept (Poster)
**Confidence:** 4

**Metareview:**

Paper proposes a method for offline imitation learning by setting up a three party game. The problem is definitely interesting to the CoRL community, the presentation is largely clear. The reviewers are generally enthusiastic about the method and appreciate real-world experiments. However, the main drawback is missing comparisons from [19], pointed out by two reviewers and other clarification questions.

**Post Rebuttal Update**
Authors satisfactorily compared their method to [19]. Given these evaluations, the presented work is a meaningful contribution to the growing body of work in offline imitation learning.

---

> ### Author Response · Authors · 2022-08-25
> **Response to Area Chair Fv1e**
>
> We thank the area chair for the valuable comment and suggestion. We have revised our paper based on the comments from the reviewers, and uploaded an updated version of the paper in "Revision Summary and the Revised Version of Paper". The detailed discussion regarding the distinctions of our work and DWBC please refer to "Response to All".